# The Effect of Plant Water Status on the Chemical Composition of Pistachio Nuts (*Pistacia vera* L. Cultivar Bianca)

**Adele Amico Roxas** [1,*]**, Giulia Marino** [1]**, Giuseppe Avellone** [2]**, Tiziano Caruso** [3] **and Francesco Paolo Marra** [3] 

[1] Department of Plant Sciences, University of California, Davis, One Shields Ave., Davis, CA 95616, USA; giumarino@ucanr.edu

[2] Department of Molecular and Biomolecular Science and Technology, University of Palermo, Via Archirafi 32, 90123 Palermo, Italy; beppe.avellone@unipa.it

[3] Department of Agricultural, Food and Forest Sciences, University of Palermo, Viale delle Scienze Ed. 4, 90128 Palermo, Italy; tiziano.caruso@unipa.it (T.C.); francescopaolo.marra@unipa.it (F.P.M.)

* Correspondence: adamicoroxas@ucdavis.edu; Tel.: +1-530-591-2240

**Abstract:** Pistachio nuts are worldwide appreciated for their chemical and organoleptic profiles. There are several studies on the influence of irrigation on pistachio productivity, whereas there are little available data on the influence on nut quality. In this study we characterized some qualitative traits of pistachio nuts cultivar Bianca in Mediterranean environment and how plant water status affected them. Water status had a positive and significant influence on the chlorophylls content, nuts from less stressed trees showed higher values of chlorophyll a (14.7 mg/100 g) and b (21.1 mg/100 g) compared than more stressed trees (9.3 and 11.5 mg/100 g for a and b, respectively). Solid phase microextraction technique in headspace followed by gas chromatography/mass spectrometry (HS-SPME GC/MS) identified seventeen different compounds with terpenes being the major class of volatiles; the most abundant were $\alpha$-Pinene (range 26.2–35 µg/g), D-Limonene (2.8–3.3 µg/g), 2-Carene (1.8–3 µg/g) and $\beta$-Myrcene (0.6–1.4 µg/g). Overall, we found higher level of terpenes in less stressed trees and for $\alpha$-Pinene and $\beta$-Myrcene differences were significant. The fatty acid composition analysis revealed oleic acid (70.1–71.1%), linoleic acid (13.5–14.4%) and palmitic acid (9.6–9.8%) as the most abundant compounds, but tree water status did not influence their concentration. Overall, the data reported proved that supplemental irrigation contributes to increase pistachio nut quality.

**Keywords:** nut quality; irrigation; stem water potential; fatty acids; chlorophylls; volatile compounds

## 1. Introduction

Pistachio (*Pistacia vera* L.) is a nut tree species native to Western Asia and Asia Minor that can be cultivated under drought and saline conditions [1]. Iran, USA, Turkey, China and Syria are the world's main pistachio producers [2]. In Italy, pistachios have been historically produced in Sicily, on lava rich soils in a niche area around Mount Etna [3]. More recently, cultivation also expanded to the center of the Island [4]. Even though the Italian production represents less than 1% of the worldwide production (3864 tons) [2], it is characterized by very high quality and sold at higher prices than pistachio nuts from other countries [2,5–7]. In particular, it is globally appreciated for the its intense green color and high nutrients content as well as its organoleptic properties [5,8]. Other factors that make Italian pistachios unique are the different processing methods and final use. In California, the second worldwide producer after Iran [2], pistachios are primarily sold as snack food, roasted with or without salt [9], while Sicilian pistachios, locally known as "green gold", are mostly sold as fresh

product, usually dried, and used in the confectionary industries as ingredients for cakes, pastries, ham, mortadella and ice creams [8]. During the roasting process, several thermal and chemical reactions occur that change some aromatic characteristics of the nuts. Specifically, it has been reported that roasted pistachios have higher concentrations of volatiles compounds than the raw ones [10]; moreover, chlorophylls are susceptible to heat and during the roasting process are converted to pheophytins and pyropheophytins, causing the color to alter [11].

Recently, the popularity of pistachio nuts increased due to its health benefits [2,5,8,12,13]. From the nutritive point of view, pistachio fat content is about 50–60% of total nut weight depending on the cultivars [14]. The fatty acid composition is dominated mostly by oleic acid, followed by linoleic and palmitic acids [14–17]. Furthermore, the organoleptic quality plays an important role for pistachio industry from the commercial point of view. The organoleptic quality of the pistachio nut is essentially correlated to the volatile profile which determines its flavor [18], and to the chlorophyll content which affects the internal kernel color [8]. Regarding the volatile composition, the terpenes class has already been identified as main group in the cultivar Kerman and in several other Iranian and Turkish varieties, with α-Pinene, D-Limonene and β-Myrcene as main constituents of the terpenes class [10,17–21].

Giovannini and Condorelli [22] were the first to study the metabolism of the chloroplast pigments in pistachio nuts and reported the presence of chlorophylls a and b, β-carotene and lutein. Agar [15] analyzed the variability of chlorophylls levels in different pistachio cultivars from several countries and found the highest chlorophylls content in pistachio nuts from Italy. Bellomo and Fallico [23] also reported similar results. These studies showed a clear effect of the genotype on pistachio nuts' quality and sensory profile, but they do not deepen the effect of management practices on nuts' quality.

In the Mediterranean Basin, pistachios have been traditionally cultivated under rainfed conditions [1,3,24], but in recent years the number of irrigated pistachio orchards have increased because of the well-claimed positive influence of irrigation on orchard productivity [25–27]. Despite the large literature showing correlation between water status and productivity, no clear information is available on the influence of trees' water status on pistachio nuts' quality. The only two studies that are available show no effect of irrigation on chlorophylls content but a negative impact on volatile composition [17,21]. However, the irrigation treatments applied in these studies were aiming to match full crop evapotranspiration. In most of the Mediterranean areas where pistachio is cultivated water is scarce, and orchards are often rainfed. We recently proved that supplemental irrigation can significantly improve yield of rainfed orchard by 30% [26]. However, if this increase in yield happens at the expense of the yield quality, it could impact the economical sustainability of the cultivation that is built on the possibility to have a high-quality product sold at a higher price in the market.

For all the above reasons, we conducted a two-year study to characterize the impact of supplemental irrigation on the qualitative characteristics (chlorophylls composition, volatile compounds and fatty acids composition) of pistachio nuts cultivar Bianca. The main aim of the study was to understand if improving the plant water status would impact overall nut quality.

## 2. Materials and Methods

### 2.1. Experimental Site and Plant Material

The experiment was carried out during two consecutive years (2014 and 2015) in a commercial pistachio orchard in Southern Italy (Caltanissetta, 37°26'02" N, 14°03'12" E; altitude 360 m). Nuts were collected from nine 30-year-old trees, cultivar Bianca, grafted on *Pistacia terebinthus* rootstock.

The climate at the experiment site is typically Mediterranean, characterized by dry and hot summers and mild winters with irregular rainfalls mostly distributed outside a four/five-months summer drought period [28]. Trees were subjected to three different irrigation treatments: rainfed—no water applied ($T_0$), 50 mm ($T_1$) and 100 mm ($T_2$) of water supplied during the growing season (from May until August). The experimental design was a complete block. Three blocks of nine trees distributed on three adjacent rows were replicated three times for each irrigation treatment. Three trees for each block

were selected for their uniformity in the central row for a total of nine trees per treatment. Water was supplied to the trees by two pressure compensating integral driplines per row with emitters (1.6 and 3.5 L h$^{-1}$) spaced 80 cm along the pipe. Daily climatic data were acquired from a public weather station (SIAS—Servizio Informativo Agrometeorologico Siciliano), 0.6 km far from the experimental site. Cumulated precipitations were 421.2 mm in 2014 and 682.6 mm in 2015. In both years, rainfalls were concentrated during fall–winter months. Taking into consideration the rainfall during the growing season and the amount of water supplied, the total amount of water received by plants was 103.6 mm in $T_1$ and 158.5 mm in $T_2$ plants in 2014, while it was 123.5 mm in $T_1$ and 175.2 mm in $T_2$ plants in 2015. Rainfed trees received through rainfalls 57.4 mm in 2014 and 80 mm in 2015. The average temperature during the growing season followed very similar pattern in the two experimental years. Starting from April (when the average temperature was 13.8 °C and 13.6 °C in 2014 and 2015, respectively), the temperature increased gradually (average temperature in June was 22.7 °C in 2014 and 22.5 °C in 2015), reaching the maximum in August (average 26.7 °C and 26.5 °C in 2014 and 2015, respectively). At the harvest in September, the average temperature was 24.2 °C in 2014 and 23.6 °C in 2015.

### 2.2. Plants Water Status

The water status of the trees was monitored by measuring midday steam water potential ($\Psi_{SWP}$) with a Scholander pressure chamber (PMS Instrument Co., Corvallis, OR, USA). Measurements were taken biweekly from May to the end of August, on two fully expanded shaded leaves per tree (eighteen samples in total) selected in bearing branches and positioned in the middle of the canopy. One hour before measurement, the leaves were covered with transparent film and aluminum foil in order to stop transpiration and equilibrate the leaf with stem water potential [29]. Pistachio is a resinous plant, so a piece of blotting paper was used to determine the end point distinguishing turpentine exudation from xylem water [25,30]. $\Psi_{SWP}$ data were used to calculate the cumulated stem water potential (cumulated $\Psi_{SWP}$), according to the method reported by Gucci [31] (Equation(1)):

$$\text{cumulated } \Psi_{SWP} = \frac{(\Psi_{SWP\ t1} + \Psi_{SWP\ t2}) \times (t2 - t1)}{2}. \tag{1}$$

where $\Psi_{SWPt1}$ and $\Psi_{SWPt2}$ are the measured $\Psi_{SWP}$ values in two consecutive dates and t1 and t2 represent the dates when measurements were taken.

Using the final cumulative value, trees were gathered into three groups:

- Water stress level 0: cumulated $\Psi_{SWP} > -130$ MPa
- Water stress level 1: $-140$ MPa $\leq$ cumulated $\Psi_{SWP} \leq -130$ MPa
- Water stress level 2: cumulated $\Psi_{SWP} < -140$ MPa

### 2.3. Biometric Parameters

At the beginning of the growing season in both years, we randomly selected two branchlets per tree (eighteen branchlets in total). Fruits were harvested on September 10th 2014 and on September 14th 2015. The fresh weight of in-shell nuts and kernels was measured soon afterwards. Fruit samples were then deshelled, peeled and dried in a ventilated oven at 40 °C until constant weight. The dry weight of the samples was measured at the end of this process. The yield of each treatment was evaluated in term of number of nuts per branchlet.

### 2.4. Chlorophylls Content

Chlorophylls content was determined on dried kernels harvested in both years. Four randomly selected kernels per each tree (twelve kernels per treatment) were deprived of the violet tegument and grinded with a blender (SterilmixerPbi Brand); 0.25 g of powder wasmixed in 10 mL flasks with 5 mL of N,N-Dimethylformamide [32,33]. Flasks covered with aluminium foil to avoid light degradation of

pigments were kept in the refrigerator for 72 hours at 4 °C. Chlorophylls content was determined using a spectrophotometer (Cary 50 Scan. Varian, Inc.) following Moran [32] calculations (Equations (2)–(4)):

$$\text{chlorophyll a} = 12.64*A664 + 2.99*A647 \qquad (2)$$

$$\text{chlorophyll b} = -5.6*A664 + 23.26*A647 \qquad (3)$$

$$\text{chlorophyll total} = 7.04*A664 + 20.27*A647 \qquad (4)$$

where $A_{664}$ is the absorbance at 664 nm and $A_{647}$ is the absorbance at 647 nm.

### 2.5. Volatiles Composition

The samples harvested in 2015 were subjected to volatile composition analysis. The identification of the volatile compounds was performed by a headspace solid phase microextraction technique (SPME) followed by gas chromatography/mass spectrometry (HS-SPME GC/MS). Four fresh samples for each tree (thirty-six samples total) were deprived of the violet tegument. Kernels (around 4.5 g each samples) were transferred into vials with pierceable silicone rubber septa coated with polytetrafluoroethylene (PTFE). n-esanolo-d13 was used as internal standard (Cambridge Isotope Laboratories, Inc. USA) at a concentration of 4.52 ppm. Vials were covered with aluminum foil to avoid the light degradation and stored in the oven at 50 °C over night. Then, the vials were transferred in water bath at 50 °C with the SPME fiber inserted in the head space and were exposed for two hours (adsorption time). To analyze the volatile composition of pistachios, we used the fiber DVB/CAR/PDMS 50/30 μm (Supelco, Bellefonte, PA, USA) that has been reported as the best solution to extract the highest number of compounds in pistachios [34]. Then the absorption was performed by headspace mode. After two hours, the fiber was introduced in the gas chromatograph. Chromatograph separation was performed with a 30 m long and 0.25 mm inner diameter-fused silica Supelcowax column (Supelco, Bellafonte, PA, USA). The fiber was maintained for 2 min at 40 °C and then for 22 min at 220 °C. The instrumental parameters of GS/MS were as follows: sink temperature 250 °C, full scan, mass range 35–350 dalton, absorption time 20 min (1 min splitless), column flow 1 mL/min, transfer line 280 °C and ionization EI (+).

Identification of the compounds was based on the comparison of the obtained spectra with those of the Wiley7 and NIST MS Search 2005 mass spectral libraries. Quantification was carried out calculating the area of peak of each component, using the method ICIS of Xcalibur V 1.4 software (thermo electron). The relative amounts of volatiles (semiquantitative analysis) were obtained by multiplying the area ratio of the target compound/internal standard by the concentration (μg/L) of the internal standard [35].

### 2.6. Fatty Acids Composition

The fresh kernels harvested in 2015 were accurately grounded with a homogenizer. Then samples of 50 g were extracted with 200 mL n-hexane in a screw-cap flask in the dark at room temperature (around 20 °C) for 30 min with the aid of ultrasound to facilitate homogenization of the kernels. The solvent was then separated by settling and the residue was extracted again with 200 mL of fresh n-hexane. A total of three extractions were carried out (3 × 200 mL); further extractions did not allow recovery of appreciable quantities of oil. The combined extracts were added with 5 g of anhydrous $Na_2SO_4$ and then n-hexane was filtered. The solvent was removed under vacuum at room temperature; the obtained oil was flushed with a steam of dry nitrogen to volatilize the solvent residues. All the oil samples were stored at −5 °C before the clean-up. The fatty acids composition of the oil was determined via GC-MS. The methyl esters of fatty acids (FAMEs) were obtained from the triglycerides by transesterification reaction [36]. The qualitative and quantitative assessment of fatty acids was carried out according to the official European Commission Regulation EEC methodology (2568/91).

### 2.7. Statistical Analysis

The effects of water stress level were analyzed as a two-way analysis of variance (ANOVA) using the Systat package (SYSTAT Software Inc., Chicago, IL, USA) and with irrigation treatment and year as fixed factors. When appropriate, Tukey's test at $p < 0.05$ was used to separate means.

## 3. Results

The number of nuts per branchlet was not affected by the water status, as shown in Table 1. Contrarily, we found a significant yearly variation. In 2015, the number of nuts per branchlet was 60% higher than the previous year (28.2 and 61.5 in 2014 and 2015, respectively).

**Table 1.** Effect of water stress levels and the year on the following biometric parameters: numbers of fruits per branchlet (n°), fresh weight (F.W., g) and dry weight (D.W., g) of nuts and kernels.

| Main Factors | Nuts/Branchlet (n°) | Nut F.W. (g) | Kernel F.W. (g) | Nut D.W. (g) | Kernel D.W. (g) |
|---|---|---|---|---|---|
| **Stress Level** | | | | | |
| 0 | 38.5 ± 5.59 [ns] | 2.2 ± 0.07 [ns] | 0.8 ± 0.08 [ns] | 1.0 ± 0.21 [ns] | 0.4 ± 0.03 [ns] |
| 1 | 49.7 ± 7.01 | 2.0 ± 0.08 | 0.8 ± 0.11 | 0.8 ± 0.22 | 0.5 ± 0.06 |
| 2 | 46.4 ± 7.01 | 2.0 ± 0.08 | 0.8 ± 0.13 | 0.8 ± 0.21 | 0.6 ± 0.06 |
| **Year** | | | | | |
| 2014 | 28.2 ± 5.19 [***] | 2.5 ± 0.17 [***] | 0.8 ± 0.08 [***] | 1.2 ± 0.05 [***] | 0.5 ± 0.03 [***] |
| 2015 | 61.5 ± 5.46 | 1.53 ± 0.20 | 0.7 ± 0.03 | 0.5 ± 0.08 | 0.41 ± 0.05 |
| **Interaction** | | | | | |
| Stress level × Year | ns | ns | ns | ns | ns |

Values are express as mean ± SD (n = 6). Values in the same column followed by different lettersindicate significant differences (Tukey's test $p \leq 0.05$); ns, not significant ($p > 0.05$). [***] $p < 0.001$.

Irrigation had no effect on the other measured parameters (fresh and dry weight of nuts and kernels), as shown in Table 1. Differences between years were also significant for these parameters. In 2015 we found a significantly lower fresh and dry weight of both nuts and kernels (38 and 60% less in nuts and 18 and 16% less in kernels, respectively).

The cumulated water stress level had a positive and significant influence on chlorophylls content, as shown in Table 2. Less stressed trees (level 0) showed significantly higher values of chlorophylls compared with the more stressed ones (level 2) (14.7 compared to 9.3 mg/100 g for chlorophyll a, 21.1 compared to 11.5 mg/100 g for chlorophyll b and 35.5 compared to 21 mg/100 g for total chlorophyll, respectively). No differences in chlorophylls content were found between years.

**Table 2.** Effect of water stress levels and year on chlorophylls content (mg/100 g). Chl a = chlorophyll a, Chl b = chlorophyll b and Chl tot = chlorophyll total.

| Main Factors | Chl a (mg/100 g) | Chl b (mg/100 g) | Chl tot (mg/100 g) |
|---|---|---|---|
| **Stress level** | | | |
| 0 | 14.7 ± 1.81 [b] | 21.1 ± 3.86 [b] | 35.8 ± 5.51 [b] |
| 1 | 11.7 ± 1.74 [ab] | 16.3 ± 3.22 [ab] | 28.0 ± 4.92 [ab] |
| 2 | 9.3 ± 1.60 [a] | 11.5 ± 3.25 [a] | 21.0 ± 4.75 [a] |
| **Year** | | | |
| 2014 | 12.0 ± 2.73 [ns] | 17.6 ± 4.93 [ns] | 29.6 ± 7.64 [ns] |
| 2015 | 11.6 ± 3.16 | 14.4 ± 5.20 | 26.0 ± 8.65 |
| **Interaction** | | | |
| Stress level × Year | ns | ns | ns |

Values are express as mean ± SD (n = 3). Values in the same column followed by different letters (a–b) indicate significant differences (Tukey's test $p \leq 0.05$);ns, not significant ($p > 0.05$).

Volatile compound composition revealed that terpenes was the major class of volatile profile. The most abundant compounds were α-Pinene (ranging 26.2–35.0 µg/g), D-Limonene (2.8–3.3 µg/g), 2-Carene (1.8–3.0 µg/g) and β-Myrcene (0.6–1.4 µg/g) (Table 3).

**Table 3.** Volatile compounds (µg/g) of pistachio nuts, aromatic description and odor threshold (ppb) affected by different levels of water stress.

| Volatile Compounds | Level 0 (µg/g) | Level 1 (µg/g) | Level 2 (µg/g) | Aromatic Descriptor | Odor Treshold (ppb) |
|---|---|---|---|---|---|
| **Terpenes** | | | | | |
| α-Pinene | 35.0 ± 4.39 [b] | 33.1 ± 1.98 [ab] | 26.2 ± 1.88 [a] | pine, turpentine | 6 |
| Camphene | 0.8 ± 0.39 [ns] | 0.7 ± 0.64 | 0.7 ± 0.09 | vanilla | - |
| β-Pinene | 0.7 ± 0.34 [ns] | 0.5 ± 0.28 | 0.5 ± 0.05 | pine, resin, turpentine | 140 |
| β-Myrcene | 1.4 ± 0.45 [b] | 0.8 ± 0.03 [ab] | 0.6 ± 0.14 [a] | balsamic, must, spice | 13 |
| 3-Carene | 0.6 ± 0.46 [ns] | 0.4 ± 0.28 | 0.3 ± 0.11 | lemon, resin | 77 |
| p-Cymene | 0.4 ± 0.44 [ns] | 0.3 ± 0.24 | 0.3 ± 0.02 | solvent, gasoline, citrus | 11 |
| D-limonene | 3.3 ± 1.01 [ns] | 2.8 ± 0.45 | 2.8 ± 0.68 | citrus, mint | 10 |
| 2-Carene | 3.0 ± 3.10 [ns] | 2.5 ± 1.44 | 1.8 ± 0.40 | sweet, pine, cedar | - |

For the aromatic descriptors were used Flavornet and NIST (National Institute of Standards & Technology) websites. For the odor threshold values was used the LRI & Odour Database Values are express as mean ± SD (n = 3). Values in the same row followed by different letters (a-b) indicate significant differences (Tukey's test $p \leq 0.05$); ns, not significant ($p > 0.05$).

In addition, the volatile analysis detected alcohol, pyrroles, esters and hydrocarbons present, for a total of seventeen volatile compounds (data not shown). Statistical analysis revealed a significant effect of plant water status on two compounds, α-Pinene and β-Myrcene. Specifically, α-Pinene significantly increased from 26.1 µg/g in most stressed trees (level 2) to 34.9 µg/g in less stressed ones (level 0), while β-Myrcene increased from 0.6 µg/g to 1.4 µg/g. Overall, we always measured a higher amount of terpenes in less stressed trees compared to the more stressed ones.

Finally, the identified fatty acids in this study were oleic, linoleic, palmitic, myristic, palmitoleic, stearic, vaccenic, linolenic, arachidic and gondoic acids, as shown in Table 4. The most abundant were the monounsaturated oleic acid (ranging from 70.1 to 71.5%), the polyunsaturated linoleic acid (from 13.2 to 14.4%) and the saturated palmitic acid (from 9.6 to 9.8%). Water status had no influence on any of the acids measured.

**Table 4.** Fatty acid composition (percentage in oil, %) of kernels affected by different levels of water stress.

| Fatty Acids | Level 0 (%) | Level 1 (%) | Level 2 (%) |
|---|---|---|---|
| Myristic acid C14:0 | 0.1 ± 0.00 [ns] | 0.1 ± 0.00 | 0.1 ± 0.00 |
| Palmitic acid C16:0 | 9.8 ± 0.07 [ns] | 9.6 ± 0.22 | 9.6 ± 0.10 |
| Palmitoleic acid C16:1 | 0.8 ± 0.06 [ns] | 0.8 ± 0.06 | 0.8 ± 0.06 |
| Stearic acid C18:0 | 1.9 ± 0.32 [ns] | 2.1 ± 0.37 | 2.2 ± 0.18 |
| Oleic acid C18:1 | 70.1 ± 1.83 [ns] | 71.5 ± 1.96 | 71.1 ± 0.73 |
| Vaccenic acid C18:1 | 2.1 ± 0.03 [ns] | 2.0 ± 0.13 | 2.0 ± 0.11 |
| Linoleic acid C18:2 | 14.4 ± 2.23 [ns] | 13.2 ± 1.98 | 13.5 ± 0.85 |
| Linolenic acid C18:2 | 0.2 ± 0.01 [ns] | 0.2 ± 0.00 | 0.2 ± 0.03 |
| Arachidic acid C20:0 | 0.2 ± 0.01 [ns] | 0.2 ± 0.01 | 0.2 ± 0.01 |
| Gondoic acid 20:1 | 0.4 ± 0.07 [ns] | 0.4 ± 0.05 | 0.4 ± 0.04 |

Values are express as mean ± SD (n = 3). ns, not significant ($p > 0.05$).

## 4. Discussion

In our experiment, tree water status did not affect the biometric parameters measured in the branchlets (number of nuts per branchlet, fresh and dry weight of nuts and kernels); similar results have been reported by previous studies on pistachio [17,25,26,37]. In this species, irrigation generally affects yield by increasing the percentages of bearing and nonbearing shoots within a canopy rather

than increasing the number of fruit per branch or fruits' weight [38]. The differences observed between 2014 and 2015 in nut biometric parameters presented in Table 1 are most likely associated with the alternate bearing behavior typical of pistachio [25,26], since temperature patterns were very similar in both years.

Water status had a positive and significant effect on chlorophylls content, as shown in Table 2. In general, water stress reduces leaf chlorophyll content [39–41] by the result of damage to chloroplasts caused by active oxygen species [42]. Despite the large amount of literature on the effect of stress on leaf chlorophyll content, only one study has been conducted on the influence of the tree's water status on pistachio kernels' chlorophylls content, and no significant effect of irrigation on this parameter was found [17]. A possible explanation for this difference may be the lower stress level reached by the plants in Carbonell-Barrachina's study [17], in which $\Psi_{SWP}$ never reached values below −1.5 MPa. Previous studies have demonstrated that only when $\Psi_{SWP}$ decreases below −1.5 MPa pistachio trees can be considered physiologically stressed showing reduction of gas exchanges [26,43,44]. In addition, the period when stress was applied may have played a significant role in determining the different response observed in these studies. In fact, Carbonell-Barrachina et al. [17] applied stress only during stage II of fruit development, when embryos are not yet fully growing [27] and pistachio trees are generally characterized by high drought resistance due to changes in the elastic modulus and a higher degree of osmotic adjustment [45]. In our study, the highest level of water stress was reached in Stage III of fruit development (data not shown) when embryos are rapidly expanding [27]. Furthermore, pistachio trees are more sensitive to water stress in this stage with respect to Stage II [27]. An additional reason for the lower chlorophyll content under water stress observed in our study can be associated to the effect of irrigation on splitting percentage. In our experiment, we did not measure splitting percentage, but Marino [26] found, in the same experimental orchard of this study, a significantly higher splitting percentage (56%) in rainfed trees in comparison with irrigated ones (33%). This effect was consistent among years. It is possible the higher splitting allows light penetration to the kernel that degrades chlorophylls, subsequently leading to the lower chlorophyll content in stressed trees.

Volatile compounds were mostly composed by terpenes class and analysis revealed the main presence of α-Pinene, D-limonene, 2-Carene and β-Myrcene, as shown in Table 3, confirming what previous studies have been reported on pistachio nuts [10,18,21]. In details, Kendirci and Onogur [18] in nuts of four Turkish cultivars (Uzun, Kirmizi, Halebi and SiirtveOhadi) found α-Pinene (ranging from 15.53 to 48.57%), D-limonene (from 3.15 to 30.04%) and β-Myrcene (from 3.50 to 8.95%) as main components. Hojjati [10] reported α-Pinene and D-Limonene as the major compounds in fresh and roasted pistachio nuts in three different Iranian varieties (Ahmad Aghaei, Akbari and Kaleghouchi). Specifically, they reported in fresh pistachio nuts α-Pinene values ranging from 1.6 to 2.05 mg/kg while in roasted ones from 2.52 to 3.96 mg/kg, and D-Limonene values ranging from 0.34 to 1.63 mg/kg in fresh nuts, while from 3.41 to 6.59 mg/kg in roasted ones. In the cultivar Kerman, Dragull [19] reported D-Limonene as the main volatile compound in nut essential oil (80–85%) and in leaf essential oil (78–81%); similar results were reported by Roitman [20] (67% in nuts and 53.6% in leaves, respectively). Furthermore, terpenes have been reported as dominant class in the leaves of *P. terebinthus* and *P. lentiscus* [46], and in nuts of *P. terebinthus* [47], suggesting that it might be a characteristic related to the *Pistacia* species.

We also found a positive and significant effect of plant water status on two compounds, α-Pinene and β-Myrcene, as shown in Table 3. As above-mentioned, only few studies have been conducted on the influence of water status on pistachio qualitative characteristics and similar results were found. In fact, Carbonell-Barrachina et al. [17] reported that a slight deficit irrigation (stem water potential slightly below −1.3 MPa during stage II) increased the concentration of volatile compounds in pistachio nuts; Sahan and Bozkurt [21] reported that irrigated conditions over the season had lowered the concentration of volatile compounds. The different trend found in our experiment could be explained as a result of a combination of (1) different degrees of stress and (2) the period of the season when the stress was applied, considering the high influence that the phenological stage of fruit development has

on pistachio capability to physiologically deal with water stress [25] and the strong seasonal pattern of terpenes associated with kernel maturation [48].

The identified fatty acids in this study shown in Table 4 confirm what has been previously reported by several authors in their studies on different cultivars from numerous countries [14–17]. In details, the high value of oleic acid found in our samples (70%) is slightly higher than values reported by Tsantilli [8] for cultivar Cerasola from Italy (67.86%) and cultivar Pontikis from Greece (68.33%). Arena et al. [10], in their fatty acids characterization of pistachio from different origins, reported that the amount of oleic acid is 70–72% in pistachios from Sicily (very similar to the values that we found) and 68–70% in those from Turkey and Greece, while Iranian pistachios were found to have the lowest amount (55%).

Finally, we found no effect of water status on fatty acid composition as shown in Table 4, and as previously demonstrated in pistachio [17], as well as in other crops [49,50]. Changes in fatty acid composition are generally function of genotype [51] or environmental factors (mostly temperature) [52]—both parameters were not variable in our experimental conditions.

## 5. Conclusions

The present study revealed that supplemental irrigation had a positive effect on the concentration of pigment chlorophylls a and b (an important eye appeal characteristic of Sicilian pistachio) and two volatile compounds responsible for the nutty characteristic of pine and balsamic flavors. Considering the scarcity of literature on this subject, this study represents an important step toward the clarification of mechanisms that regulate organoleptic pistachio nut quality. These results are particularly important considering the big interest of consumers and fresh market industries in healthy food and high quality products characterized by specific sensory profiles.

The first investigation on Sicilian pistachio cultivar Bianca proves that supplemental irrigation, in addition to improving yield [26], contributes to an increase of some qualitative characteristics. Results suggest the need of further research studies on this topic that take into consideration different degrees of plant water stress and different periods of stress application to fully capture the particular effect of water status on the quality traits of pistachio fruits.

**Author Contributions:** A.A.R. performed the experiment, analyzed data and wrote the paper. G.M. performed the experiment and revised the manuscript. G.A. performed the chemical analysis. T.C. revised the manuscript. F.P.M. conceived and designed the experiment, analyzed data and revised the manuscript. All authors have read and agreed to the published version of the manuscript.

**Funding:** This research received no external funding.

**Conflicts of Interest:** The authors declare no conflict of interest

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
