# Peer review of "The Effect of Plant Water Status on the Chemical Composition of Pistachio Nuts (Pistacia vera L. Cultivar Bianca)"

_agriculture, doi:10.3390/agriculture10050167_

Round 1

Reviewer 1 Report

I have examined the manuscript “Effect of Plant Water Status on Chemical and Sensory Properties of Pistachio Nuts (Pistacia vera L. cultivar Bianca)” (agriculture-788081). The study is relevant since it describes the impact of water status on quality-related minor components, as well as productivity traits. However, some aspects of the manuscript should be improved before being accepted for publication. Following is a list of my comments and suggestions to improve the paper.

Title

Line 2 - Sensory properties were not measured for this experiment. Sensory related minor components were measured. The title should be changed since it is not representative of the study.

Abstract

Line 20 – Mass spectrometer analysis is too general, please revise. Significative figures should be revised as well.

Introduction

Line 37, 47, 57 + others – Need a reference. All statements from the introduction should be backed up by a reference.

Line 41: Is “green product” referring to “fresh product”, is so please replace.

Line 50-53 – Phenolic compounds are not measured in this study, their description is irrelevant, please correct.

Line 80 – Why fat content of the pistachio nuts was not considered in this experiment? Given that part of the health benefits of pistachio nuts are related to its fatty acid concentration it would have been relevant to consider this parameter as well. Please indicate why this parameter was not considered.

Material and methods

Line 110 – Correct font size

Linie 123: How were nut and kernel weight measures performed?

Line 124 – Does drying temperature affect chlorophylls and/or volatile content determinations in pistachios?

Lines 132-133 – Use the same format as in Lines 113-114.

Line 136 – Change M for m

Line 135-138 – Improve description, SPME is not a mode, it is an analytical technique.

Line 140: How much sample was used? Was it a dried sample? Was it fresh?

Line 143: Why the samples are incubated overnight? Does this lead to the degradation of volatile compounds?

Line 154 – 156: Are results expressed on a wet or dry basis? This might modify the conclusions of the experiment. How’s the moisture content of the pistachio nuts?

Results

Tables 1-4: Revise significative figures, how was replication handled? The error includes variations among trees, or it is a representation of the analytical methodology error? Please clarify. The relative errors for fresh and dry weights are very low (~3%) for a natural product as pistachios.

Discussion

Line 224 – The authors claim that irrigation caused a significant effect on chlorophyll concentration. Are these differences relevant from a sensory standpoint?

Line 282 – Was temperature measured during this experiment? If so, please present the data and include it in the discussion.

Line 269 – …considering the high… please revise the sentence.

Line 279 – Fatty acid concentration is expressed in relative area, not in concentration. Therefore, even though the overall synthesis of lipids was affected by the treatments, no difference would have been detected. A measure of absolute concentration is required in this case. To this end, the measurement of the total oil content in the nut would have been helpful. Please indicate if these measurements are available and if so, include them on the manuscript.

Conclusions

Line 288 – Chlorophylls and volatile compounds are only related to the sensory characteristics of pistachios. Please remove the nutraceutical from the sentence.

Reviewer 2 Report

As far as I'm concerned, the paper is clear and easy to read.

I suggest that the authors check misprints.

The title is not entirely appropriate but as the authors
have determined volatile components that contribute
to aroma, I did not correct it.

Line 41: maybe "green" should be changed into "fresh"

Lines 113: font of formula

Lines 109-110: check the font

Lines 132-133: check the font of formula

Line 136: modify the description of SPME. The word "mode" is not correct.

Round 2

Reviewer 1 Report

The authors properly addressed all the questions and made the suggested changes.